# Er-Doped Tapered Fiber Amplifier for High Peak Power Sub-ns Pulse Amplification

**Maksim M. Khudyakov [1],\***, **Andrei E. Levchenko [1]**, **Vladimir V. Velmiskin [1]**, **Konstantin K. Bobkov [1]**,
**Svetlana S. Aleshkina [1]**, **Mikhail M. Bubnov [1]**, **Mikhail V. Yashkov [2]**, **Aleksei N. Gur'yanov [1]**,
**Leonid V. Kotov [3]** and **Mikhail E. Likhachev [1]**

1. Prokhorov General Physics Institute of the Russian Academy of Sciences, Dianov Fiber Optics Research Center, 38 Vavilov Street, 119333 Moscow, Russia; a_levchenko@fo.gpi.ru (A.E.L.); vvv@fo.gpi.ru (V.V.V.); bobkov@fo.gpi.ru (K.K.B.); sv_alesh@fo.gpi.ru (S.S.A.); bubnov@fo.gpi.ru (M.M.B.); guryanov@ihps-nnov.ru (A.N.G.); likhachev@fo.gpi.ru (M.E.L.)
2. Institute of Chemistry of High Purity Substances of the Russian Academy of Sciences, 49 Tropinin Street, 603950 Nizhny Novgorod, Russia; yashkovmv@ihps-nnov.ru
3. James C. Wyant College of Optical Sciences, University of Arizona, 1630 E. University Blvd, Tucson, AZ 85721-0094, USA; leonidkotov@arizona.edu
* Correspondence: mkhudyakov@fo.gpi.ru

**Abstract:** A tapered Er-doped fiber amplifier for high peak power pulses amplification has been developed and tested. The core diameter changed from 15.8 µm (mode field diameter (MFD) 14.5 µm) to 93 µm (MFD 40 µm) along 3.7 m maintaining single-mode performance at 1555 nm (according to the $S^2$-method, the part of the power of high-order modes does not exceed 1.5%). The amplification of 0.9 ns pulses with spectral width below 0.04 nm up to a peak power above 200 kW (limited by self-phase modulation) with a slope pump-to-signal conversion efficiency of 15.6% was demonstrated.

**Keywords:** taper; Er-doped; high efficiency; high peak power; self-phase modulation; very large mode area

## 1. Introduction

Erbium-doped fiber (EDF) lasers and amplifiers emitting at the spectral range near 1.55 µm are relatively eye-safe [1]. Thus they are widely used in a variety of applications requiring light transmission through the atmosphere, such as wind doppler LIDAR (Light Detection and Ranging) [2,3], atmospheric gas concentration detection from space [4,5], and atmospheric communication. Such applications require good beam quality, high peak, and average power, which are, however, difficult tasks for EDF amplifiers to accomplish simultaneously.

The most straightforward fiber amplifier setup consisting of an EDF doped fiber core-pumped by a single-mode pump laser diode is limited in both peak and average output power. To increase the peak power, it is necessary to increase the core diameter and/or decrease the length of the active fiber. The shortening of the fiber length is mainly limited by the upconversion of $Er^{3+}$ ions due to clustering at high concentrations, which leads to a decrease in the pump-to-signal conversion efficiency (PCE) and limits pump absorption. An increase in the core diameter is limited by a deterioration of the beam quality due to a growth of the number of guided modes. The average power is usually increased by utilizing multimode high power laser diode modules for cladding-pumping double-clad (DC) active fibers. In this case, a higher core diameter is beneficial because of the direct correlation between pump absorption and core/cladding ratio. Thus, the main way to solve all three issues is to find a way to increase core size while maintaining single-mode operation.

The simplest solution is to increase the core diameter and simultaneously lower the numerical aperture (NA). This results in large mode area (LMA) DC fiber amplifiers. The

most common type of such amplifiers is Er-Yb codoped pedestal-based LMA DC fiber amplifiers, which can achieve a peak power of ~40 kW for ns pulse amplification with a slope PCE of ~5% in 5 ns pulses [6]. In terms of peak power scaling, such amplifiers are inferior to Yb-free LMA DC EDF amplifiers [7–9], which achieved a peak power of 84 kW in 2 ns pulses with a PCE of 8.4% [9]. Noticeable improvement of the PCE was demonstrated using a combination of Yb-free EDF and Er-Yb doped fiber in the single amplifier [10]. In this case, 37 kW peak power was achieved with 37% PCE [11]. The core diameter with this approach is limited to 30–35 μm due to an increase in bending losses with decreasing NA, which makes larger core low NA fibers impractical.

Single-mode operation of multimode fibers can be ensured by a higher differential amplification of a fundamental mode, which is achieved by bending the fiber [12]. It is worth noting that the increase of the fiber core diameter increases the normalized frequency and decreases the maximum bend diameter necessary to keep the single-mode operation. A smaller bend diameter leads to a decrease in the mode field diameter (MFD), limiting this approach's applicability [13].

Core pumping of Er-doped fibers is another approach to creating a higher differential gain for the fundamental mode in highly multimode fibers by the selective excitation of the fundamental mode by the pump and the signal light. Thus, the fundamental mode at the signal wavelength has the highest overlap with the pump mode and the highest gain [14]. Nevertheless, such schemes rely on non-standard high-brightness single mode pump sources (typically—Raman fiber laser, operated near 1480 nm and pumped by Yb-doped fiber laser [14]) and utilize custom-made high power wavelength division multiplexors (to combine pump and signal), making them more complex and reducing efficiency compared to the LMA DC EDF setups [15].

Another way to increase the core diameter without degrading beam quality is to introduce new waveguiding elements to the first cladding of the fiber, which facilitates resonant coupling of higher-order modes (HOMs) away from the core [16–18]. However, the resonant nature of this approach means that they are effective for a small spectral range and sensitive to the optical parameters of the core and cladding elements. It means that the practical implementation of single-mode amplifiers based on such fibers becomes laborious.

One more promising solution was developed for Yb-doped fibers operated in another spectral region—near 1 μm. In this case, the record peak power for an all-fiber amplifier with diffraction-limited beam quality was achieved using tapered fiber geometry [19]. In a tapered fiber, its diameter increases along the fiber length to a value several times larger than the initial diameter. The thin part of the tapered fiber has a small core diameter necessary for a single-mode operation with a second mode cut-off wavelength lower than the operational wavelength. The slow adiabatic increase in diameter along the length of the fiber ensures zero energy transfer between the fundamental mode and the HOMs. Tapered EDFs (T-EDFs) also demonstrated record or close the record values of peak power for a variety of pulse widths [20–22]. The highest peak power for ns pulse amplification was 107 kW [22]. However, the high erbium ion concentration in such T-EDFs reduces their PCE to the value of 3–4% due to clustering.

The goal of this work was an optimization of the T-EDF parameters to increase the PCE and maintain a high threshold of nonlinear effects at the same time.

## 2. T-EDF Design and Optimization

The T-EDF preform in this work was manufactured by the MCVD method. The glass matrix chosen in this work was $Al_2O_3$-F-$SiO_2$ with a high fluorine concentration. This choice was dictated by a higher absorption peak at 976 nm at a fixed Er concentration (compared to $P_2O_5$-$Al_2O_3$-$SiO_2$ glass matrix) and its position compatible with standard wavelength-stabilized multimode pump diodes [8]. The core was doped with $2 \cdot 10^{25}$ m$^{-3}$ Er$^{3+}$ ions.

One of the most critical factors determining the PCE of the cladding-pumped DC EDFs is the core/cladding ratio. The more the ratio, the higher the rate of pump absorption

and the shorter the length of active fiber. The main factor limiting PCE in EDF is erbium ion clustering and the associated upconversion, which acts as an unbleachable loss. As a result, shortening the active fiber reduces the influence of an unbleachable loss and increases the overall PCE. The maximum core/clad ratio in the case of tapered fibers is limited by the single-mode light propagation requirement at the thin end.

To reduce the bending sensitivity of the T-EDF, which was a problem in previous works [20–22], the NA of the core was increased to 0.09 (see Figure 1a). The utilization of a W-shaped refractive index profile (RIP), realized by a thin fluorine-doped layer around the core, allows achieving single-mode operation at 1550 nm in a core having a diameter of up to 16 μm. A further increase in the core diameter, at which it can still operate in the single-mode regime, can be achieved by additional bending of the thin end of the T-EDF with a small diameter [12].

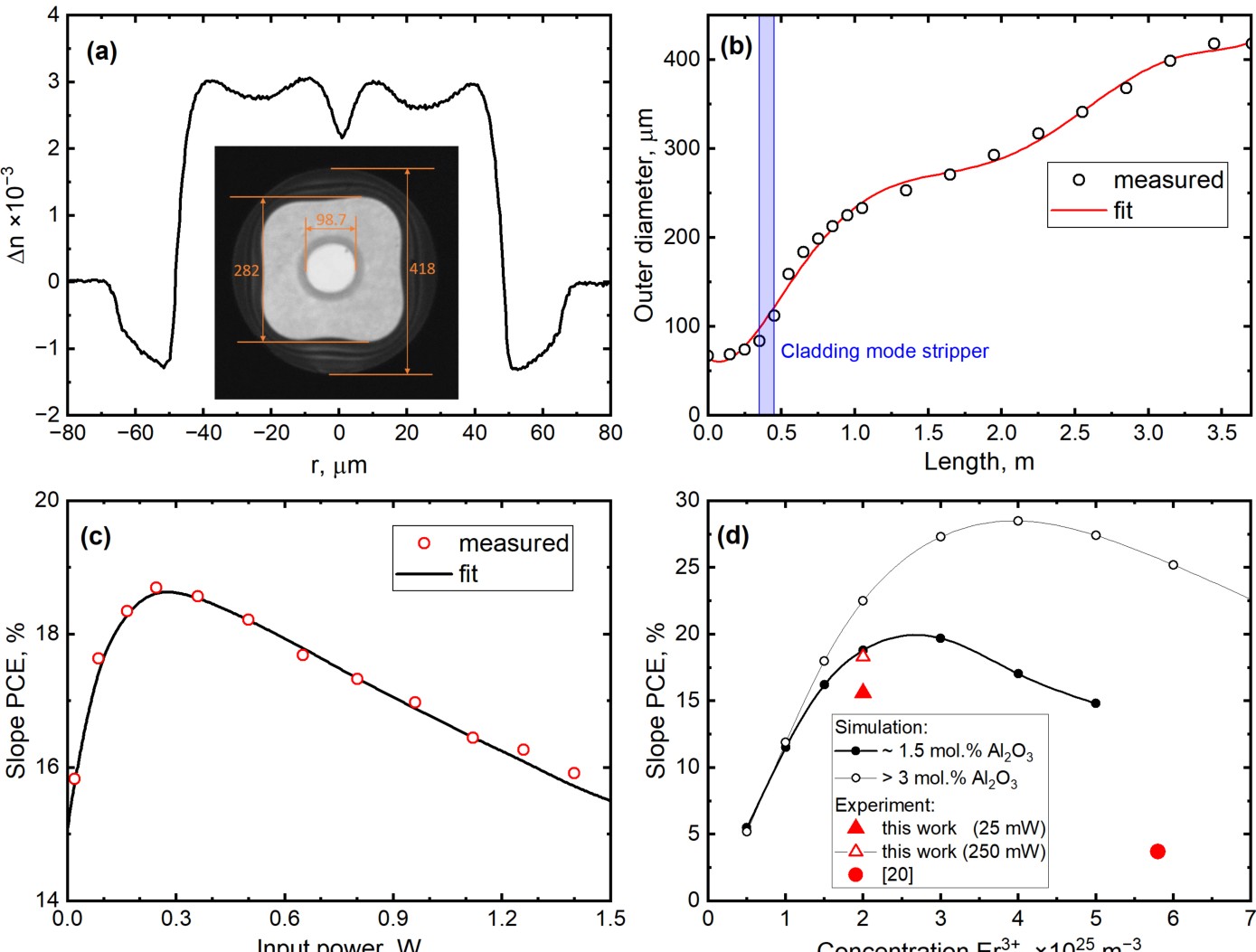

**Figure 1.** (**a**) RIP of the T-EDF; inset: photo of the thick end facet; (**b**) diameter distribution along the T-EDF length (red curve was used in the simulation), the blue bar represents the location of the cladding mode stripper produces on the T-EDF; (**c**) signal saturation curve of the T-EDF (the circles correspond to measured values and the black curve is an approximation); (**d**) calculated slope PCE of the T-EDF with clustering curves for different alumina concentrations and experimental data. In (**c,d**) the signal wavelength was 1555 nm and the pump wavelength was 976 nm.

The second limiting factor for the core/cladding ratio is a minimal outer diameter of 80 μm necessary for the developed fiber to be compatible with standard splicing and cleaving equipment. Using a second fluorine-doped reflective cladding around the pure

silica cladding for guiding pump light allowed us to further increase the core/cladding ratio. For this purpose, we developed a technique for overcoating a silica preform by a thick layer of fluorine-doped silica glass. The ratio between diameters of the fluorine-doped cladding and the pure-silica cladding was as high as 1.4. In addition, the silica preform was square-shaped to break the circular symmetry of the cladding pump modes (see inset in Figure 1a). Thus, the average diameter of the first inner cladding of the developed fiber with an outer diameter of 80 μm was 59.8 μm. The core diameter was 18.9 μm, and the core/first cladding ratio was equal to 0.316.

The T-EDF has been drawn from the fabricated preform. The outer diameter of the T-EDF changed from 67 μm to 420 μm over a length of 3.7 m (see Figure 1b). The geometry was determined by the drawing process and could not be significantly altered. The core diameter changed from 15.8 μm to 98.7 μm. The calculated MFD at 1555 nm for the thin end was 14.5 μm, and the estimated cut-off wavelength was 1540 nm. Thus, it was completely single-mode at the signal wavelength. The calculated MFD of the thick end was 56 μm. The disproportionally smaller increase in MFD is caused by uneven RIP in the center of the core.

We measured the PCE of the T-EDF in a counter-pumped scheme, where the CW signal was launched into the thin end of the fiber and the pump was coupled by a pair of lenses into the thick end facet of the T-EDF. The end facet was polished at an angle of 7° to prevent Fresnel reflection. The dichroic mirror was used to separate the pump and signal. The setup scheme is presented in the lower part of Figure 2 (the seed is replaced by a CW source emitting at 1560 nm). The measurement of the slope PCE dependence on the input signal power is presented in Figure 1c. The maximal slope PCE was 18.7% for the input power of 0.25 W.

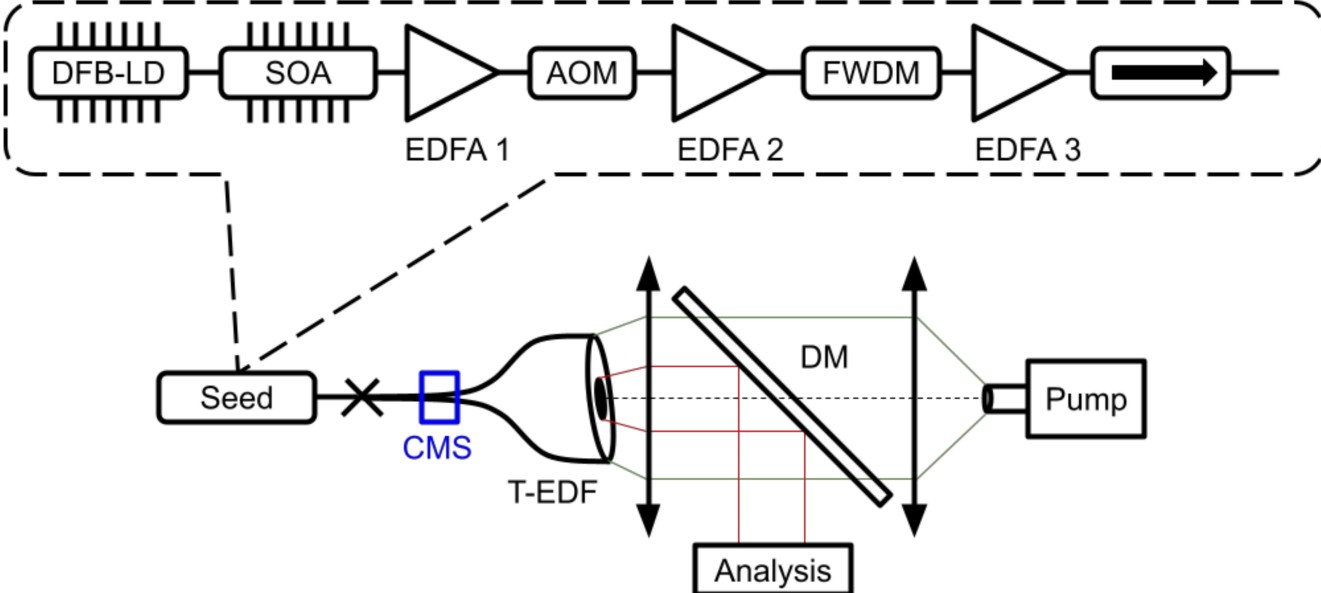

**Figure 2.** Scheme of the setup of a sub-ns pulse amplifier based on the T-EDF.

To evaluate the chosen parameter of the T-EDF, we conducted a numerical simulation of the slope PCE dependence on the erbium ion concentration in the T-EDF. The pair-induced quenching model described in [23] was used for the numerical modeling. The dependences of the number of paired ions on the erbium concentration for different glass hosts were taken from [15].To properly model the T-EDF amplifier, it is important to consider the vignetting effect [24,25] caused by the increase of the pump NA while the fiber diameter decreases as light travels from the thick end to the thin end. Consequently, the pump light aperture can exceed the maximum allowable NA maintained by the first reflecting cladding of the fiber, after which the pump having propagation angles greater

than the angle of total internal reflection leaks into the second cladding and at a further point into the air through the lateral surface of the fiber. The pump light is considered to travel along the fiber axis, and the NA intensity distribution is equal to that of a semiconductor pump diode with an NA of 0.15 measured in [26]. The clustering dependence on erbium concentration for the $Al_2O_3$-F-$SiO_2$ matrix with a high fluorine concentration has not been studied yet. However, it is reasonable to assume that it will be between the values obtained for $Al_2O_3$-$SiO_2$ matrices with low and high alumina concentrations [15]. Both dependencies were used in the numerical simulation, the result of which is presented in Figure 1d.

It is worth noting that the PCE of the T-EDF described in [20–22] was only 3–4% at an erbium ion concentration of $6 \cdot 10^{25}$ m$^{-3}$. Considering that it had an $Al_2O_3$-$P_2O_5$-$SiO_2$ glass matrix and taking into account the result of [8], it can be assumed that its PCE would be comparable to that of the T-EDF with an $Al_2O_3$-F-$SiO_2$ glass matrix. In addition, from our previous study [27], one would expect it to be between the curves for high and low alumina doped matrices. Therefore, the low PCE for T-EDF in [20–22] is likely due to some overlooked fact. For example, the pump NA can be significantly different from what was assumed due to the angled thick end facet. Nevertheless, the realization of the same or higher erbium concentration in the core of the T-EDF is inadvisable due to already obtained low PCE. According to the simulation results, the selected erbium ion concentration is in the optimal range of $(2–3) \cdot 10^{25}$ m$^{-3}$, resulting in peak cladding small-signal absorption of 1.8 dB/m at 976 nm. Higher concentration results in higher clustering and decreases efficiency. At a lower concentration of erbium ions, the pump absorption from the cladding becomes smaller and decreases the PCE. In addition, a larger amount of unabsorbed pump power can cause damage to the T-EDF at the leakage point [25]. Thus, the chosen Er concentration in the fabricated T-EDF seems to be a little lower than the optimal one.

## 3. Experimental Setup

The experimental setup is presented in Figure 2. A distributed feedback laser diode (DFB-LD) with a center wavelength of 1553.45 nm, an output power of 1 mW, and a linewidth of 2 MHz was used as a seed source. Its radiation was coupled into the semiconductor optical amplifier (SOA), driven by a custom-designed electrical circuit to produce 880 ps pulses with a repetition rate of 10 kHz and a peak power of 10 mW (average power 0.1 μW). The pulses were first amplified by a core-pumped EDF amplifier (EDFA 1) with an acousto-optical modulator (AOM) afterward to filter out amplified spontaneous emission (ASE) caused by low input power, resulting in an average power of ~0.1 mW (10 W peak). Another core-pumped EDF amplifier (EDFA 2) with a filtering wavelength division multiplexor (FWDM) with a 0.6 nm FWHM was used to amplify the signal to an average power of ~3 mW (300 W peak). Further amplification with small-core amplifiers resulted in the onset of the four-wave mixing (FWM), so in the third stage, a short LMA cladding-pumped EDF amplifier (EDFA 3) was utilized to increase the average power to 25 mW (2.5 kW peak). The resulting signal was launched through an isolator into the thin end of the T-EDF.

The thick end of the T-EDF was polished at an angle of 7° to prevent the Fresnel reflection into the core. The thick end of the T-EDF was pumped by a multimode wavelength-stabilized at 976 nm laser diode with an output power of up to 27 W through two lenses with an 8 mm focal length and a maximum NA of 0.5. A dichroic mirror (DM) was used to separate the output signal from the incoming pump.

It is worth noting that the MFD of the thin end of the T-EDF was lower than that of the LMA fiber used in the EDFA 3. The portion of the input signal, which was lost due to MFD mismatch, can still propagate in the first cladding. Due to the large core/cladding ratio cladding modes can have non-zero intensity in the core, which can cause excitation of HOMs in the core and degradation of the beam quality. To eliminate this problem, we fabricated a cladding mode stripper (CMS) around the leakage point of the T-EDF (see blue line in Figure 1b).

## 4. Result of Sub-ns Pulse Amplification

The results of sub-ns pulse amplification are presented in Figure 3. We used an integrating photodetector to determine the portion of the ASE after T-EDF [19]. It was less than the measurement accuracy (~1% of total power) at all pump powers. The slope PCE of the T-EDF was 15.6%, which is in line with our previous measurements in Figure 1c for ~20 mW of input power. Thus, the maximum pulse energy of 313 μJ was achieved at a pump power of 26.7 W (see insert in Figure 3a). This allowed us to calculate maximum achieved peak power using the waveform measured with a fast stroboscopic oscilloscope (see Figure 3c) equal to 361 kW. However, this peak power level is well above the T-EDF nonlinearity threshold in this setup.

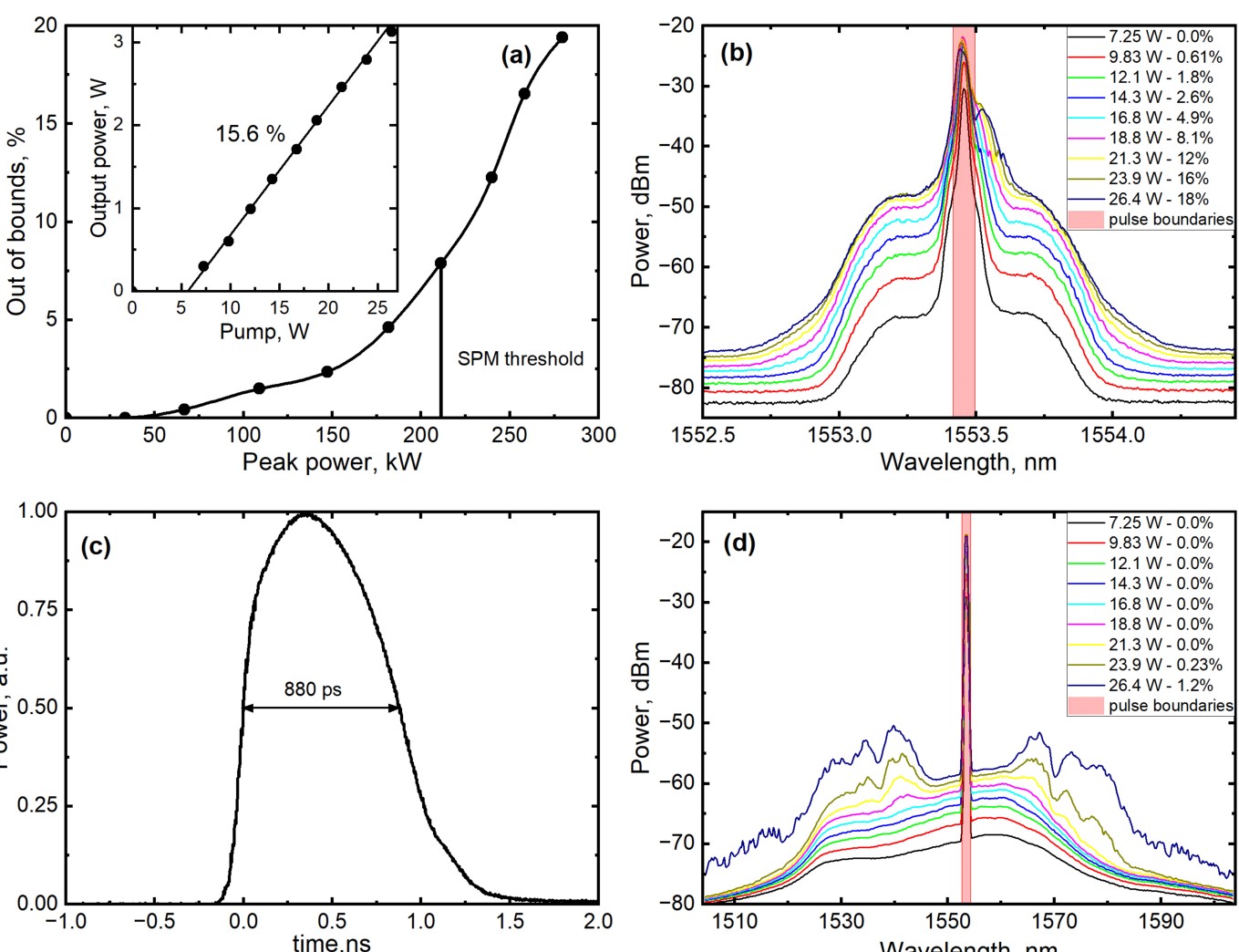

**Figure 3.** Results of ns pulses amplification in the T-EDF. (**a**) The out of bounds part of the output power as shown in (**b**) and (**d**) vs. peak power; inset: output power vs. peak power; (**c**) the waveform of the output pulse measured with 50 GHz stroboscopic oscilloscope; (**b**,**d**) output spectrum at a resolution of 0.02 nm (**b**) and 1 nm (**d**). The red rectangle indicates the part of the spectrum which is considered "in-pulse." Anything outside the red rectangle is considered to be outside of pulse boundaries for energy and peak power calculations. Pulse boundaries widths are 0.04 nm (**b**) and 2 nm (**d**).

Self-phase modulation (SPM) and FWM are the main nonlinear effects that limit the peak power of sub-ns pulses. Therefore, we used an optical spectrum analyzer (OSA) to assess their thresholds. The effect of self-phase modulation is best seen at the highest spectral resolution (see Figure 3b). Exceeding the SPM threshold results in broadening of the pulse spectrum near the operating wavelength. Since the time-limited spectral bandwidth of sub-ns pulses is much lower than the highest spectral resolution of our OSA,

we chose spectral bandwidth of 0.04 nm ($2\times$ spectral resolution), which corresponds to a $-20$ dB bandwidth of the original pulse. We calculated the fraction of the output power outside the pulse boundaries for each pump power and subtracted the value at the lowest pump power, which corresponds to the spectral noise in the input signal (this did not affect the calculation of pulse energy and peak power). At pump powers higher than 18 W, the pulse bandwidth increases dramatically; thus, we consider that to be the SPM threshold in our experiment. At 18.8 W of pump power, the output power was 2.06 W, which corresponds to a pulse energy of 183 μJ and peak power of 211 kW at the SPM threshold. Another nonlinear effect that we have observed in the system is FWM. FWM causes new peaks to appear at significantly different from the main peak wavelengths. Figure 3d shows that at pump powers over 18.8 W, additional peaks arise in the output spectra. However, according to the integral measurements similar to the ones for the SPM but with a spectral bandwidth of 2 nm. the intensity of the process is an order of magnitude lower. Thus, the main limiting factor for sub-ns peak power scaling of the T-EDF is the SPM with a threshold of ~200 kW.

## 5. Beam Quality Measurements

To maintain single-mode operation and good output beam quality, the RIP of the tapered fiber must be relatively smooth. As the diameter of the core rises, the feature size of any imperfection in the RIP increases, and as it approaches the signal wavelength, it can affect light propagation in the tapered fiber core. Unfortunately, the T-EDF in this work has a large dip on the fiber axis. This dip is a feature of aluminosilicate glass preform highly doped with fluorine and it is caused by the evaporation of highly volatile compounds $AlF_3$ during preform consolidation [28]. This should adversely affect the beam quality.

To evaluate the spatial beam characteristics of the manufactured T-EDF, we carried out several measurements with a CW input signal with a central wavelength of 1560 nm and a pump power of 17 W (output power ~2.5 W). The $M^2$ value of the output beam was 1.34/1.31 for the $x$ and $y$ axes correspondingly (see Figure 4a), which indicates near-diffraction-limited beam quality and single-mode operation. To further verify the single-mode performance, we measured multipath interference using the $S^2$ measurement technique with single-mode scanning optical fiber [25,29]. As shown in Figure 4c, the multipath interference (MPI) plot exhibits several peaks, but the only meaningful peak is the one with a delay of 2.88 ps and a $-18.7$ dB MPI corresponding to the $LP_{11}$ mode (see Figure 4e, where retrieved intensity and phase correspond to the $LP_{11}$ mode). This confirms the single-mode operation of the T-EDF.

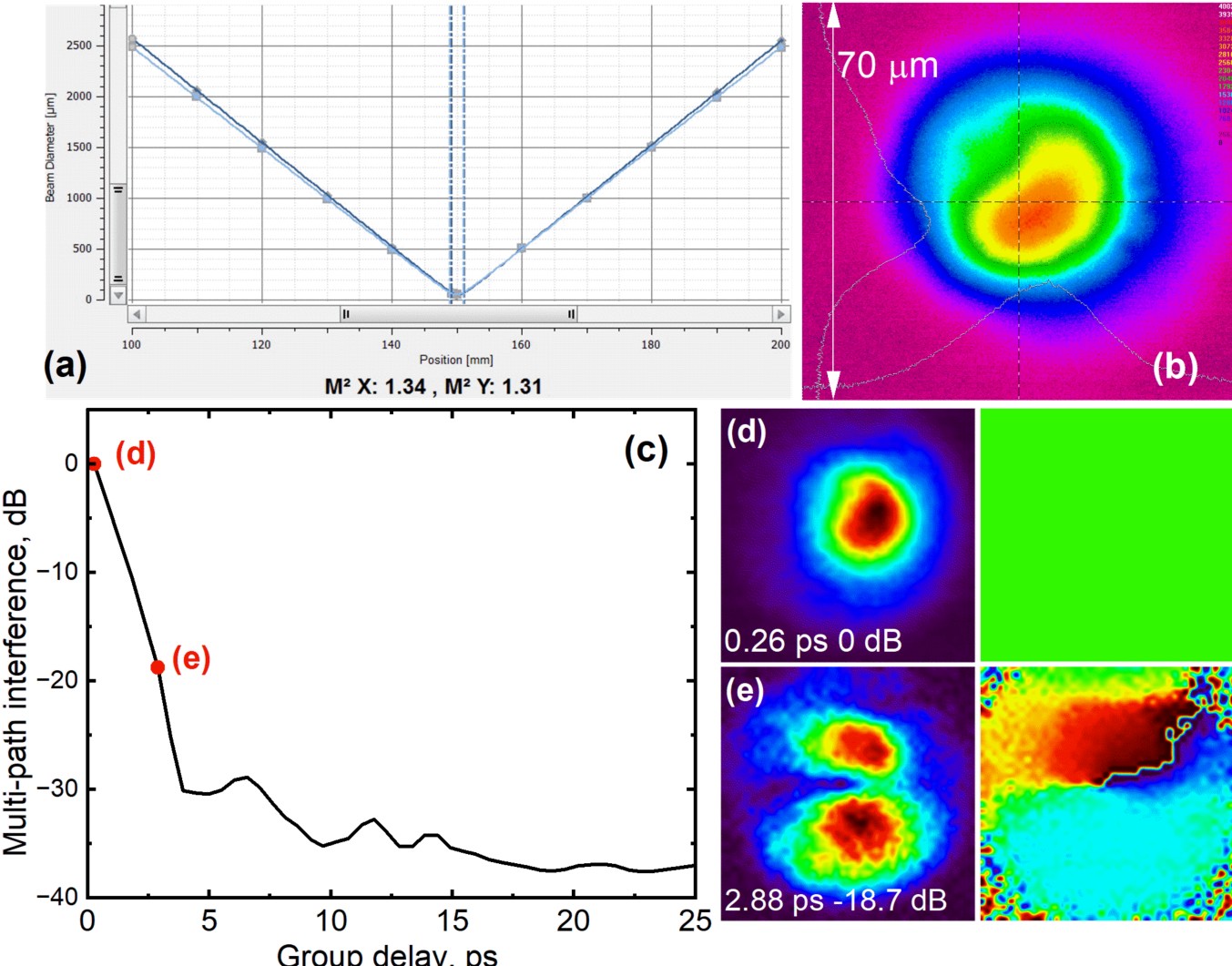

**Figure 4.** Beam quality measurement in the T-EDF (conducted with CW input). (**a**) $M^2$ measurement; (**b**) focused facet image; (**c**) $S^2$ measurement; (**d**,**e**) are retrieved mode intensities (left, color scale normalized from 0 to 1) and phases (right, color scale from $-\pi$ to $\pi$) at corresponding points of the $S^2$ plot.

Apart from increasing the value of $M^2$, the most significant effect of the central dip in the T-EDF RIP is the distortion of the output mode field, which can be seen in Figure 4b. Due to the uneven RIP and the bending of the T-EDF (even with a large diameter of 30 cm), the mode intensity peak shifted away from the center of the core. This is the reason for the imperfect $M^2$ values despite the single-mode performance. In addition, this "squeezing" of the mode to the edge of the core results in a decrease of the MFD. Direct measurement of the MFD from the focused image gives a value of ~40 μm, which is lower than the value of 56 μm calculated from the RIP (core diameter is ~99 μm).

## 6. Discussion and Conclusions

As was shown in the Introduction the ordinary fibers with diameters of the core and the cladding which are constant along the fiber length do not have much room for improvement—core diameter increase in such fibers is limited by the requirement to have a reasonably low bend loss and a single-mode operation regime. Even utilization of novel combined Er/Er-Yb amplifier design [10,11], which allowed us to noticeably improve the PCE, does not result in an increase of peak power in the single-frequency pulses above the record one obtained previously [30].

A completely different situation was demonstrated in this work for the case of tapered Er-doped fiber. The T-EDF developed in this work demonstrates the possibility of achieving high peak powers over 200 kW in sub-ns pulses with a relatively high slope PCE of 15.6% with near-diffraction-limited beam quality and ultra-narrow linewidth of 0.04 nm. Achieved peak power is more than twice as high as the one achieved in cladding-pumped DC LMA step-index regular fibers [9] (84 kW vs. 211 kW) with almost twice higher slope PCE (15.6% vs. 8.4%). In addition, this work demonstrates the possibility of increasing the efficiency of the T-EDFs compared to previous works [20–22] (five times slope PCE increase from 3% to 15.6% with 1.5 times higher peak power). Note that an even higher peak power of 361 kW was achieved at the cost of a wider output pulse spectrum (99% of power within 1 nm spectral width).

The achieved PCE is a record high for T-EDFs. It is 5 times higher than in the previous works [20–22] with 1.5 times higher peak power for the same type of pulses [22]. Moreover, according to our calculations presented in Figure 1d the chosen erbium concentration is not optimal and further PCE increase is possible.

The MFD at the thick end of the fiber was diminished due to the uneven RIP. It is possible to achieve higher MFD with smoother RIP, which will also increase the nonlinearity threshold of the T-EDF. Moreover, optimization of the drawing process to increase the thick end diameter to the thin end diameter ratio will result in higher MFD. Furthermore, our calculations revealed some unidentified factors, which limited the PCE of the T-EDF at a high Er concentration, so further optimization of the concentration will be possible if such factors are eliminated. The realization of smooth RIP will increase both the output MFD and the nonlinearity thresholds in the T-EDF.

In conclusion, the demonstrated T-EDF already shows the record high peak power and PCE. Moreover, keeping in mind non-optimal parameters of the current tapered fiber we could conclude that T-EDFs geometry offers unprecedented peak power scaling potential for lasers operated in the 1.55 μm range.

**Author Contributions:** Conceptualization, M.E.L., M.M.B., and A.N.G.; methodology, M.M.K., A.E.L., V.V.V., K.K.B., S.S.A., M.V.Y., and M.E.L.; software, M.M.K. and L.V.K.; validation, M.M.K., L.V.K., K.K.B., and M.E.L.; formal analysis, L.V.K.; investigation, M.M.K.; resources, M.M.K., K.K.B., V.V.V., M.V.Y., and M.E.L.; data curation, M.M.K. and L.V.K.; writing—original draft preparation, M.M.K.; writing—review and editing, M.M.K., L.V.K., S.S.A., and M.E.L.; visualization, M.M.K. and L.V.K.; supervision, M.E.L.; project administration, M.E.L.; funding acquisition, M.E.L. All authors have read and agreed to the published version of the manuscript.

**Funding:** This research was funded with the financial support of the Ministry of Education and Science in the form of grant No. 075-15-2020-912 for the creation and development of a world-class research center "Photonics".

**Institutional Review Board Statement:** Not applicable.

**Informed Consent Statement:** Not applicable.

**Conflicts of Interest:** The authors declare no conflict of interest.

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
