# Peer review of "Er-Doped Tapered Fiber Amplifier for High Peak Power Sub-ns Pulse Amplification"

_photonics, doi:10.3390/photonics8120523_

Round 1

Reviewer 1 Report

This paper demonstrates a tapered Er-doped fiber amplifier for high peak power pulses amplification, The core diameter changed from 15.8 µm to 93 µm maintaining single-mode performance at 1555 nm, the beam quality is High and the paper is well written. Therefore, I think this paper is appropriate to be published, and I have some questions following:

  1. In the introduction line 31, the “space” is repeated, I think this may be a typo.
  2. In the line 122, ‘using a second fluorine-doped reflective cladding around the pure silica cladding allowed us to further increase the core/cladding ratio.’ Why the F-doped cladding can increase the core/cladding ratio? Can the pump light get through the F-doped cladding? I think some details should be given.
  3. More information should be given about the numerical model in Fig.1b. What is the role of the Er concentration in the model? Does it influence the PCE by changing the clustering ratio, or other mechanisms?
  4. In the line 171, “optimal range of 2—3∙1025 m-3”, the format of the symbol is wrong.
  5. What is the absorption coefficient of the taper fiber? Is the length of fiber (3.7m) decided by the absorption?

Author Response

  1. In the introduction line 31, the “space” is repeated, I think this may be a typo.

Replaced “space [4], space [5]” with “space [4], [5]”

  1. In the line 122, ‘using a second fluorine-doped reflective cladding around the pure silica cladding allowed us to further increase the core/cladding ratio.’ Why the F-doped cladding can increase the core/cladding ratio? Can the pump light get through the F-doped cladding? I think some details should be given.

Replaced “Using a second fluorine-doped reflective cladding around the pure silica cladding allowed us to further increase the core/cladding ratio” with “Using a second fluorine-doped reflective cladding around the pure silica cladding for guiding pump light allowed us to further increase the core/cladding ratio. For this purpose, we developed a technique for overcoating a silica preform by a thick layer of fluorine-doped silica glass.”

Also replaced “Consequently, the pump light aperture can exceed the maximum allowable NA maintained by the first reflecting cladding of the fiber, after which the pump having propagation angles greater than the angle of total internal reflection leaks through the lateral surface of the fiber” with “Consequently, the pump light aperture can exceed the maximum allowable NA maintained by the first reflecting cladding of the fiber, after which the pump having propagation angles greater than the angle of total internal reflection leaks into the second cladding and at a further point into the air through the lateral surface of the fiber”

  1. More information should be given about the numerical model in Fig.1b. What is the role of the Er concentration in the model? Does it influence the PCE by changing the clustering ratio, or other mechanisms?

Added “Higher concentration results in higher clustering and decreases efficiency.” at line 177.

Added “The pair-induced quenching model described in [23] was used for the numerical modeling. The dependences of the number of paired ions on the erbium concentration for different glass hosts were taken from [15].” at line 149-152

Added”23. Nilsson,J.; Jaskorzynska,B.; Blixt,P. Implications of Pair-Induced Quenching for Erbium-Doped Fiber Amplifiers, in Optical Amplifiers and Their Applications, 1993, MD19.” to references

  1. In the line 171, “optimal range of 2—3∙1025 m-3”, the format of the symbol is wrong.

Replaced “2—3∙1025 m-3” with “(2—3)∙1025 m-3

  1. What is the absorption coefficient of the taper fiber? Is the length of fiber (3.7m) decided by the absorption?

Added “The geometry was determined by the drawing process and could not be significantly altered.” at line 133.

Also changed “According to the simulation results, the selected erbium ion concentration is in the optimal range of (2—3)∙1025 m-3.” with “According to the simulation results, the selected erbium ion concentration is in the optimal range of (2—3)∙1025 m-3, resulting in peak cladding small-signal absorption of 1.8 dB/m at 976 nm. Higher concentration results in higher clustering and decreases efficiency.” at line 175-178.

Reviewer 2 Report

Please correct the typo in the title, and check for grammatical mistakes.

Starting at line 252 you discuss the "dip" in the preform being due to alumina burnout. Alumina does not burn out, in fact it collects to slightly increase the index during consolidate and collapse process. I believe this fluorine burnout. Is there an explanation for the presence of fluorine in your fiber?

Author Response

  1. Please correct the typo in the title, and check for grammatical mistakes.

We made various corrections aimed at improving English level and correcting mistakes.

Replaced “Er-Doper Tapered Fiber Amplifier for High Peak Power sub-ns Pulse Amplification” with “Er-Doped Tapered Fiber Amplifier for High Peak Power sub-ns Pulse Amplification” in the title.

  1. Starting at line 252 you discuss the "dip" in the preform being due to alumina burnout. Alumina does not burn out, in fact it collects to slightly increase the index during consolidate and collapse process. I believe this fluorine burnout. Is there an explanation for the presence of fluorine in your fiber?

Added “Unfortunately, the T-EDF in this work has a large dip on the fiber axis. This dip is a feature of aluminosilicate glass preform highly doped with fluorine and it is caused by the evaporation of highly volatile compounds AlF3 during preform consolidation [28].” At line 258-261

Added “28. Yashkov,M. V et al. Optical properties of heavily ytterbium- and fluorine-doped aluminosilicate core fibres. Quantum Electron. 2017, 47, 1099–1104” to references

Reviewer 3 Report

This paper presents a tapered Er-doped fiber (T-EDF) amplifier for high peak power pulse amplification, and their optimization of the T-EDF parameters to increase the pump-to-signal conversion efficiency (PCE) and maintain a high nonlinear threshold effect. This manuscript lacks the novelty and inadequacy of final optimized T-EDF parameters to increase the PCE and high nonlinear threshold values. Some of the major concerns are;

  1. Since achieving a high peak power based on Er-doper tapered fiber is a well-demonstrated research area in the literature, the authors need to explain clearly what are the differences and novelty of this manuscript compared to them.
  2. The main objective of this paper is to study T-EDF sustainability and maintain a high threshold of nonlinear effects. However, this manuscript lacks rigorous analysis on various nonlinear effects, their threshold levels, and their dependence on tapered fiber length.
  3. Page 6, Self-phase modulation (SPM), which is the smallest threshold level and Four-wave mixing (FWM) are the main nonlinear effects that limit the peak power of sub-ns pulses.”. What about the stimulated Raman/Brillouin scattering? How the nonlinear threshold depends on the length of the tapered fiber?.
  4. What is the cut-off wavelength of T-EDF?
  5. A conclusion table showing optimized T-EDF parameters and PCE values are helpful.
  6. The abbreviations need to be defined, for instance, NA,

Author Response

  1. Since achieving a high peak power based on Er-doper tapered fiber is a well-demonstrated research area in the literature, the authors need to explain clearly what are the differences and novelty of this manuscript compared to them.

In the introduction we clearly state that tapered fiber geometry allowed to achieve high peak powers in Yb-doped fibers. However, all realized tapered Er-doped fibers to date have demonstrated low pump conversion efficiency (~3%). Thus: “The goal of this work was an optimization of the T-EDF parameters to increase the PCE and maintain a high threshold of nonlinear effects at the same time.”

We suggest that starting a discussion of tapered fiber geometry by discussing Yb-doped fiber, which acts in another spectral range (1 mm) was a bit confusing. However, it is reasonable as the most remarkable demonstration of the advantages of tapered fiber geometry was done with Yb-doped tapered fiber. To outline that we talk about the different spectral range we replaced

”In the spectral region near 1 µm, the record peak power results were achieved using an all-fiber amplifier with diffraction-limited beam quality by utilizing the tapered fiber geometry [19].”

with “One more promising solution was developed for Yb-doped fibers operated in another spectral region - near 1 µm. In this case, the record peak power for an all-fiber amplifier with diffraction-limited beam quality was achieved using tapered fiber geometry [19].” at line 80

  1. The main objective of this paper is to study T-EDF sustainability and maintain a high threshold of nonlinear effects. However, this manuscript lacks rigorous analysis on various nonlinear effects, their threshold levels, and their dependence on tapered fiber length.

Length dependency was addressed earlier in the answer to the first reviewer: “The geometry was determined by the drawing process and could not be significantly altered.” The demonstrated nonlinear effects (SPM and FWM) are the ones with the lowest thresholds for the chosen pulse parameters. We could investigate the thresholds of SBS or Raman, however, that would require different pulse parameters and is beyond the scope of the paper.

  1. Page 6, Self-phase modulation (SPM), which is the smallest threshold level and Four-wave mixing (FWM) are the main nonlinear effects that limit the peak power of sub-ns pulses.”. What about the stimulated Raman/Brillouin scattering? How the nonlinear threshold depends on the length of the tapered fiber?

See answer to 2.

  1. What is the cut-off wavelength of T-EDF?

“The core diameter changed from 15.8 µm to 98.7 µm. The calculated MFD at 1555 nm for the thin end was 14.5 µm, and the estimated cut-off wavelength was 1540 nm.” line  133. It is important to note that cut-off wavelength changes with core size and is only sensible for a fixed fiber diameter.

  1. A conclusion table showing optimized T-EDF parameters and PCE values is helpful.

We consider the table unnecessary for only one tapered fiber presented.

  1. The abbreviations need to be defined, for instance, NA

Replaced “The simplest solution is to increase the core diameter and simultaneously lower the NA.” with “The simplest solution is to increase the core diameter and simultaneously lower the numerical aperture (NA)” at line 47

Round 2

Reviewer 3 Report

The authors addressed all comments and concerns. The manuscript has been improved than the previous version. I still suggest improving the English language.